# Comparative Study of Malocclusions between Cancer Patients and Healthy Peers

**DOI:** 10.3390/ijerph19074045

**Published:** 2022-03-29

**Authors:** Patrycja Proc, Joanna Szczepanska, Anna Herud, Malgorzata Zubowska, Wojciech Fendler, Monika Lukomska-Szymanska, Wojciech Mlynarski

**Affiliations:** 1Department of Pediatric Dentistry, Medical University of Lodz, Pomorska 251, 92-213 Lodz, Poland; joanna.szczepanska@umed.lodz.pl; 2Department of Orthodontics, Medical University of Lodz, Pomorska 251, 92-213 Lodz, Poland; anna.herud@umed.lodz.pl; 3Departments of Pediatrics, Diabetology, Endocrinology and Nephrology, Medical University of Lodz, Sporna 36/50, 91-738 Lodz, Poland; malgorzata.zubowska@umed.lodz.pl; 4Department of Radiation Oncology, Dana-Farber Cancer Institute, Boston, MA 02215, USA; wojciech.fendler@umed.lodz.pl; 5Department of Biostatistics and Translational Medicine, Medical University of Lodz, Mazowiecka 15, 92-215 Lodz, Poland; 6Department of General Dentistry, Medical University of Lodz, Pomorska 251, 92-213 Lodz, Poland; monika.lukomska-szymanska@umed.lodz.pl; 7Department of Pediatrics, Oncology & Hematology, Medical University of Lodz, Sporna 36/50, 91-738 Lodz, Poland; wojciech.mlynarski@umed.lodz.pl

**Keywords:** childhood cancer survivors, malocclusion, dental age

## Abstract

Background. There is lack of data related to dental occlusion among children cured from cancer. The aim of our study was to compare the prevalence of malocclusion in cancer survivors and in healthy peers. Methods. A cross-sectional study was conducted on 225 children aged between 4 and 18 years, including 75 cancer survivors, and 150 sex and age-matched controls. All patients were orthodontically examined and malocclusion traits were recorded. In the cancer group, 75 panoramic radiographs were used to evaluate the prevalence of dental anomalies and dental age using the Demirjian scale. Data were analyzed by univariate statistical analysis with *p*-values *p* < 0.05 considered as statistically significant. Results. Malocclusion was found in 49 (65.33%) cancer survivors and 99 (65.56%) controls (*p* > 0.05). The cancer group demonstrated significantly higher likelihood of crossbite (*p* < 0.01) and malalignment of teeth (*p* = 0.031). The healthy controls were more likely to demonstrate open bite (*p* = 0.038). Cancer patients with posterior crossbite (*p* = 0.023) or dental malalignment had a more advanced dental age (*p* = 0.022). Survivors with crossbite had more teeth with short roots (*p* = 0.016). Those who were older when they started their cancer therapy were more likely to suffer from tooth disturbances (*p* = 0.019). Conclusion. Oncological treatment can alter the development of occlusion in cancer patients.

## 1. Introduction

Occlusion is defined as the way the upper and lower teeth intercuspate between each other in all mandibular positions and movements [1]. It is, however, rare to identify an “ideal occlusion” described in relation to the three planes of space. In 1899, Angle first clearly described the relationship between permanent upper and lower first molars as indicators of normal occlusion [1]. According to Angle, the normal occlusion exists when the mesiobuccal cusp of the upper first molar occludes with the buccal groove of the lower first molar, this is Class I. Class II is recognized when the mesiobuccal cusp of the maxillary first molar occludes anterior to the buccal groove of the mandibular first molar. Class III when mesiobuccal cusp of the upper first molar falls posterior to the buccal groove of the lower first molar. Despite the fact that more than 100 years have passed since the definition of classes according to Angle, its assumptions are still valid in the field of anterior–posterior positioning of the jaw bones [1]. 

The Angle classes did not cover all diagnostic problems in orthodontics, so in 1972, Andrew defined the six keys to normal occlusion as molar relationship, correct crown angulation and inclination, absence of rotations, tight proximal contacts, and flat occlusal plane [2]. Availability of new diagnostic methods led to continuous improvement of concept of “ideal occlusion” but nowadays the clinicians focus more on facial proportions and the impact of the dentition on the facial appearance than on the jaw relationship itself [3]. 

The development of occlusion is a fairly dynamic process, especially in patients during the period of tooth replacement, and even after reaching maturity, constant changes take place [4]. However, the occlusion norms are generally similar in primary and permanent dentition. The upper teeth overlap the lower teeth, the upper incisors overlap the lowers by no more than a half of their crown height, one upper tooth contacts with two lower ones (except the lower central incisors and the upper last erupted molars), the dental arches are symmetric with coincident midlines. In primary dentition, the dental arches are semicircular and the occlusal plane is flat. Second deciduous molars lie in the vertical plane, canines occlude in Class I. Some children have anthropoid spaces in the anterior segment of the teeth. With the eruption of the first permanent molars, the shape of the dental arches changes into semielliptical in the maxilla and parabolic in the mandible. The occlusal plane creates the curve of Spee. The canines and the first permanent molars stay in Class I position [4].

A malocclusion is the lack of features that characterize a correct occlusion or the lack of acceptance of occlusion by the patient. By the Dental Practice Board, UK, the malocclusion has been defined as meaning “an abnormal occlusion in which teeth are not in a normal position in relation to adjacent teeth in the same jaw and/or the opposing teeth when the jaws are closed”. However, this is not always negative and is sometimes described by orthodontists as “an appreciable deviation from ideal occlusion” [5]. 

While anatomical or physiological discrepancies may have a various etiologies, it is agreed that all include genetic or environmental factors. Bad oral health habits, such as thumb/limb sucking, tongue thrusting or habitual mouth breathing are considered as particularly important [6], and dental diseases such as caries, pulpal and periapical lesions, dental trauma, or anomalies in deciduous or permanent dentitions may also play a role [7]. In patients without specific indications, the first orthodontic follow-up is recommended at the age of 7. However, some early-stage oral dysfunctions can interfere with young child bone development and normal dental relationships, leading to malocclusion [8].

Changes in occlusion may also be related to general diseases in children during their developmental stages. An orthodontically relevant orofacial manifestation termed *dysgnathia*, described as skeletal maldevelopments of the upper or lower jaw, was found in over 25% of patients with so-called rare diseases (RD) [9]. Over 17% of patients with RD demonstrated an altered number of teeth, with most demonstrating hypodontia. On the other hand, severe malocclusion may also involve the overgrowth of jaw bones, presented in medical states such as cherubism or acromegaly [10,11]. 

Childhood cancer survivors often suffer from various dental anomalies, such as misshaped tooth roots, missing or microdontic teeth, as well as various other dental complications—including dental caries or hypoplasia [12,13]. Cancer survivors sometimes present a significantly different dental age to healthy children at a similar chronological age [14]. It is well known fact that the dental complications may influence occlusion in healthy children [15,16,17]. This must also apply to children who have had cancer, but no recent articles on the subject are available.

Therefore, the aim of this study was to determine whether cancer patients are more likely to demonstrate malocclusion than their healthy peers.

## 2. Materials and Methods

### 2.1. Study Group Description

Cancer patients were recruited from an ongoing program for the assessment of late adverse effects of anticancer treatment in survivors of childhood cancers, held in the Department of Pediatrics, Oncology and Hematology, Medical University of Lodz. Detailed information on the recruitment of participants and the inclusion and exclusion criteria are provided in the diagram (Figure 1).

Although 109 cancer survivors were initially welcomed to take part in dental examinations, 34 did not meet the inclusion criteria or refused to participate. The children in the study group had been treated for: acute lymphoblastic leukemia (ALL), Wilms tumor, neuroblastoma, rhabdomyosarcoma (RMS), brain tumor, hepatoblastoma, acute non-lymphoblastic leukemia (AN-LL), B cell non-Hodgkin lymphoma (B-NHL), Hodgkin lymphoma (HL), peripheral primitive neuroectodermal tumor (PNET), germinal tumor, and ovarian tumor (Table 1). Dental examination was performed 6 to 155 months (average 4.9 years) after the cessation of anticancer treatment, with the youngest patient being 47 months old (four years) and the oldest being 215 months (18 years). 

The control group consisted of initially 296 pupils from randomly selected classes from a local kindergarten and school (Figure 1). These control patients were recruited during a time corresponding to the recruitment of the cancer survivor group. Seventy eight healthy controls were excluded as they did not meet the criteria. Then, the cancer patients were matched in a 1:2 ratio with 218 healthy controls, using the propensity score procedure described in the statistical analysis section. This resulted in a group of 225 children and adolescents (75 cancer patients and 150 controls).

Ethical approval for the study (IRB No. RNN/37/13/KE) was provided by the Bioethics Committee of the Medical University of Lodz on 19 February 2013. Written informed consent for dental examination was obtained from both parents or guardians and patients above the age of 16.

### 2.2. Orthodontic Assessment in Both Groups

The patients were orthodontically examined in the Department of Pediatrics and Orthodontics, Medical University of Lodz between July 2013 and January 2016. Examination was performed by two examiners (A.H. and P.P.) with the aid of penlight, mouth mirror, and metal millimeter ruler. The data were collected on a paper form designed for the study based on WHO recommendations 15 [18]. Occlusal assessment was carried out with the teeth in centric occlusion i.e., in the maximum habitual intercuspation of the teeth. 

The following parameters were recorded: 1. anteroposterior relations based on the first permanent molars and canines’ position on both sides (distinction between Class I—normal occlusion, Class II—distocclusion, and Class III—mesiocclusion); 2. abnormal overjet/upper incisor protrusion (greater than 3 mm); 3. reversed overjet (four incisors in crossbite); 4. vertical relations based on overbite (distinction between deep bite with upper incisors overlapping the lower by more than 1/2 of their crown height and open bite with no contact between all four incisors for anterior and at least two opposing teeth for posterior open bite); 5. transversal relations based on the buccolingual position of the first permanent molars (distinction between posterior crossbite—recorded when a buccal cusp of an upper tooth lies lingually to the maximum height of a buccal cusp of an opposing lower tooth—and scissors bite—recorded when a lingual cusp of an upper tooth lies buccally to the maximum height of a buccal cusp of an opposing lower tooth); 6. anterior crossbite (1–3 upper incisors positioned with negative overjet—in crossbite); 7. crowding (including rotations); 8. malalignment (change in tooth inclination and angulation not associated with crowding); 9. spacing (excluding physiological diastema of less than 2 mm); 10. centerline shift (associated with mandibular displacement); 11. maxillary narrowing (manifesting with long V—shaped upper dental arch and deep palatal vault).

### 2.3. Assessment of Dental Anomalies and Dental Age in Cancer Group

Seventy five panoramic radiographs of teeth in cancer patients were analyzed by two examiners (A.H. and P.P.); in the case of discrepancies, they were discussed until the agreement was achieved. Among the anomalies recorded in the cancer patients, oligo-hypodontia (smaller number of teeth), microdontic teeth (at least half of size when compared to homonymous ones), and teeth with misshaped roots were observed (root shortage was confirmed if the root/crown length ratio was lower than 1.6, according to a simplified Hölttä Defect Index); [19]. The dental age of the cancer survivors was estimated with the Demirjian method, based on the stage of development of seven lower left permanent teeth [20]. Delta age was then calculated as the difference between dental and chronological age; delta age = dental age − chronological age (DA-CA). 

### 2.4. Assessment of Treatment-Related Variables in Cancer Group

The following characteristics related to cancer patients were analyzed: cancer type, age at diagnosis, duration of treatment (number of days), radiotherapy of the head or neck (rtx), disease relapse, and total body irradiation (TBI). The types of malocclusions were divided into groups as follows: mesio-distal changes (Class II, Class III, abnormal overjet, reversed overjet), transversal changes (anterior and posterior crossbite, scissors bite, midline shift, maxillary narrowing), vertical changes (anterior and posterior open bite, deep bite), and tooth disturbances (crowding, spacing, malalignment).

The results of orthodontic treatment in the next 5 years, until 2021, were later assessed as to whether the treatment was undertaken by oncological patients at the Orthodontics Clinic.

### 2.5. Statistical Analysis

As the continuous variables did not demonstrate a normal distribution, they were represented as medians and 25–75% ranges, and the groups compared using the Mann-Whitney test. Propensity scores were calculated using a logistic regression model for group allocation depending on age and sex to balance the impact of these potentially confounding variables. After obtaining the scores, the controls were matched in a ratio of 2 per each patient. Nominal variables were compared using the Chi-squared test with Yates’s continuity correction (if the number of patients in either group was <15) or the two-tailed Fisher’s exact test (if the number of patients in either group was lower than five). *p* levels < 0.05 were considered as statistically significant.

## 3. Results

### 3.1. Comparison of Types of Dental Occlusion in Both Groups

Malocclusion was equally common among cancer patients and healthy children (65.33 vs. 65.56%, *p* = 0.97); (Table 2). Class I was found in 68.00% of the cancer survivor group and 62.25% of the healthy control group, Class II in 28% of the cancer group and 34.4% of the control group, and Class III in 8 and 2%, respectively (*p* > 0.05). 

Cancer patients were more likely to demonstrate crossbites: anterior (13.33 vs. 3.97%, *p* = 0.0136) and posterior (18.67 vs. 4.64%, *p* = 0.0012), and malalignment of teeth (18.67 vs. 7.95%, *p* = 0.0310) when compared to their healthy peers in the control group. However, they were less likely to demonstrate anterior open bite (1.33 vs. 8.61%, *p* = 0.0387). Cancer patients also demonstrated a deep bite less often than controls, but the difference was not statistically significant (1.33 vs. 7.95%, *p* = 0.0651). 

### 3.2. Dental Anomalies and Dental Age Estimation in Cancer Patients 

Children with posterior crossbite had significantly more teeth with short roots than the other cancer survivors (*p* = 0.01); (Table 3). Among them, six patients had both anterior and posterior crossbites, four had anterior crossbite, and eight had posterior crossbite. Four patients additionally had Class II malocclusion, six patients had malalignment of teeth, and one had spacing. Twelve (66.6%) children with crossbite had at least one dental anomaly. Five children missed from one to three teeth, seven had one or two microdontic teeth, and five children had two to eight teeth with misshaped roots. 

It was also found that dental age was grossly accelerated in cancer children with posterior crossbite (*p* = 0.0235). Their delta age (DA-CA) amounted up to 1.75 years when compared to 0.8 years for cancer children without crossbite (Table 4). 

The state of the dentition was shown in the panoramic images of two cancer survivors with diagnosed crossbite and tooth anomalies (Figure 2a,b).

The next group with significantly accelerated dental age were cancer patients showing dental malalignment. Their dental age was accelerated by 1.07 years (*p* = 0.022) when compared to chronological age. Among them, four had Class II occlusion, three had posterior crossbite, and one an open bite. Most (57.1%) had at least one dental anomaly. One patient had oligodontia (missing more than six teeth), one patient missed three teeth, five patients had ten microdontic teeth (each patient with two teeth), and one had four teeth with misshaped roots. Additionally, three patients had three impacted teeth (numbers 13, 34, and 36).

No other dental anomalies were found related to the type of occlusion in cancer patients.

### 3.3. Cancer Treatment Related Variables and Subsequent Orthodontic Treatment

Among cancer patients, the occurrence of malocclusion was not associated with such variables as: cancer type (data not shown), time of oncological treatment, radiotherapy (RTX) of the head or neck, disease relapse, or total body irradiation (TBI) (*p* > 0.05); (Table 5). It was found that patients who started their therapy later in life were more likely to suffer from tooth disturbances such as malalignment, crowding, or spacing (*p* = 0.0198).

After the diagnosis of late side effects of anti-cancer treatment in patients with a history of childhood cancers as part of the program, all patients with malocclusion were offered orthodontic treatment at the Orthodontics Clinic. Eight of the patients started the treatment, six received removable appliances, one, with a brain tumor, was provided dental prosthetists, and one was provided complex treatment including fixed appliances and ortho-gnathic operation. Only one patient completed treatment within 5 years of observation with a positive result; the other seven discontinued treatment at different stages for various reasons (relapse of cancer, change of place of living, discouragement).

## 4. Discussion

There are no contemporary data available which link anti-cancer therapy with malocclusion occurrence among children and adolescents. To fill this gap, the present study focused on the prevalence of malocclusion among pediatric cancer survivors.

The prevalence of malocclusion has been found to vary worldwide, with the highest levels being seen in Africa (81%) and Europe (72%), followed by America (53%) and Asia (48%) [21]. It has also been found to increase in mixed and permanent dentition in response to environmental or genetic factors; however, the prevalence of Class II and Class III problems seems to be the most stable. Globally, the distribution in permanent dentition of Class I malocclusion was found to amount up to 74.7%, Class II malocclusion up to 19.56% and Class III up to 5.93 %, with the respective distributions of these types being 73, 23, and 4% in mixed dentition [22].

In Saudi Arabia, the most common malocclusions in order of prevalence in patients aged 5–7 years were Class I (52.8%), Class II (31.8%), Class III (15.4%), crowding (47.2%), excessive overjet (22.2%), reduced overjet (11.4%), excessive overbite (23.4%), reduced overbite (12.2%), anterior crossbite (4.8%), posterior crossbite (9.4%), and open bite (4.6%), [23]. Among 444 examined Italian adolescents, 75.5% demonstrated a Class I molar relationship, but 99% of the sample showed at least one occlusal trait, such as overjet > 3 mm (48%), overbite > 3 mm (39%), midline misalignment (32%), or crowding (30%), [24].

While no data are currently available regarding the prevalence of malocclusion in a Polish population, the most observed in a young Caucasian population was Class II (22.9%), [22]. Comparable results were obtained in the present study, as Class II problems were found in 28% of cancer patients and 34.44% of controls. The global prevalence of Class III malocclusion in Caucasians has been estimated as 5.92%; similarly, in the present study, it was observed in 8% of cancer survivors and 1.99% of control children. The second most common malocclusions in the present study were anterior and posterior cross bites, the latter being present in 18.67% of the cancer patient group and 4.64% of controls; globally, posterior crossbite affects 7.53% of Caucasians.

In general, dental occlusion is believed to be grossly altered in patients with oral cancer, mainly due to significant deterioration in the mobility and sensory function of the tongue [25]. Also tooth loss and the subsequent need of dental prosthetists were found to alter masticatory function. A large questionnaire survey on 381 American orthodontists reported that the most common dental problems in cancer survivors were malalignment of teeth, root stunting and growth, or development changes [26]. Seventy-five percent of the respondents applied orthodontic treatment modifications, including lighter forces for tooth movement and antibiotic prophylaxis. In 2% of the cancer patients, complications were so severe that orthodontic therapy could not be performed, and 15% of the patients discontinued therapy prematurely.

Interestingly, the cancer patients in our group had less problems with vertical dimension malocclusions than control children. The development of anterior open bite has been associated with a range of dysfunctions, such as using a pacifier, thumb or finger sucking, mouth breathing, as well as neuromuscular deficiency, trauma, rheumatoid disease, improper posture, and posterior discrepancy [27]. It is likely that the children with cancer might not practice such oral habits due to the mucositis and other oral pains caused by cancer therapy, which would prevent open bite formation. Also, deep bite was less often identified among cancer patients than among controls; this may be due to the accelerated dental age caused by cancer therapy, which might prevent formation of deep bite. However, these hypothesis require further studies based on larger groups of cancer survivors.

Previously, it has been found that 62.3% of childhood cancer survivors demonstrated at least one dental anomaly, such as teeth with short roots, as well as microdontic or missing teeth [12]. A few papers have also confirmed a relationship between dental anomalies and occlusion [15,16,17]. A study on the prevalence of various dental anomalies found that the incidence of tooth impaction and short or blunt roots was significantly related to the type of occlusion, i.e., Class I, II, or III [15]. The correlation between Class III malocclusion, hypodontia, and microdontia was also confirmed [16]. A study on dental records of 2052 healthy Brazilian patients found that impaction was correlated with Class I, microdontia was related to Class II division I, while impaction of teeth and ectopia were correlated with Class III [17]. No such relationships were observed in the present study, possibly due to the small sample size. The only other relationship identified in the present study was that cancer patients with posterior crossbite were more likely to demonstrate teeth with short roots.

Previously, it was also found that cancer not only affects tooth development but also disturbs the process of dental maturity [14]. In most cases, cancer therapy may accelerate dental age; however, it was found to be delayed in patients suffering from (FAP)-associated hepatoblastoma [14]. Celikoglu et al. found that orthodontic patients with skeletal malocclusions demonstrated approximately twice the mean difference between dental and chronological time than patients with Class I [28]. These were compared with the Demirjian’s scale [20] and calculated with delta dental age as the difference between mean dental age and chronological age (DA-CA).

In our study, patients with posterior crossbite tended to be more advanced in dental age when compared to other cancer patients. This is in accordance with the finding that cancer patients with posterior crossbite had more teeth with malformed roots, which, on the other hand, are found often as dental complication after cancer therapy [12]. Patients suffering from malalignment also demonstrated more advanced dental development, which could be related to the disturbance in teething and lack of space for permanent successors. It stays in line with our findings that patients who started their therapy later in life exhibit more changes such as malalignment, crowding, or spacing.

The outcome of orthodontic treatment of cancer patients was also analyzed. All the patients with malocclusion were diagnosed and referred to orthodontic treatment. However, many of the cancer patients did not dwell locally to the oncology ward and dental center, and so decided to undertake treatment nearer to their place of residence. From the group of eight patients who decided to start the treatment in our dental center, only one patient finished it with satisfactory results. This is in accordance with previous observations. In a group of ten childhood cancer survivors from Sweden, four patients did not finish orthodontic treatment and one patient demonstrated resorption of tooth roots [29]. Moreover, it has been found that many cancer survivors can have psychological trauma connected with medical treatment [30]. An investigation on the quality of life (QoL) of 40 cancer survivors before, during, and after orthodontic treatment found that male cancer survivor patients reported significantly lower QoL during the course of treatment, which was not observed in the male control group; in addition, while the outcome of orthodontic treatment, if continued, did not differ from that observed in the control patients, the patients demonstrated lower long-term stability of orthodontic treatment [31,32].

The main weakness of the present study is the limited number of children recruited and low group homogeneity due to different types of cancer. However, it took almost 3 years to recruit this number of patients, so the results obtained are unique. The problems of the poor reporting of cancer patients to such programs has already been observed, due to their medical and psychological problems [33]. To increase statistical power and reduce population heterogeneity and hence reduce a potential bias, the propensity score matching with healthy peers was performed. It was also not possible to accurately compare the data with the results of other groups, because the selection of modern data on this subject is small, old, or incomplete. The number of patients also did not allow classification by groups comparing dolichofacials and brachyfacials, which is another limitation. Therefore, checking the hypothesis as to whether oncological patients suffer less often from vertical defects and more often from transverse ones requires research on a larger group.

## 5. Conclusions

Within the limitations of the present study, it could be concluded that oncological treatment in children can alter the development of dental occlusion. The crossbite type of occlusion in cancer patients was related to the presence of teeth with short roots.

## Figures and Tables

**Figure 1 ijerph-19-04045-f001:**
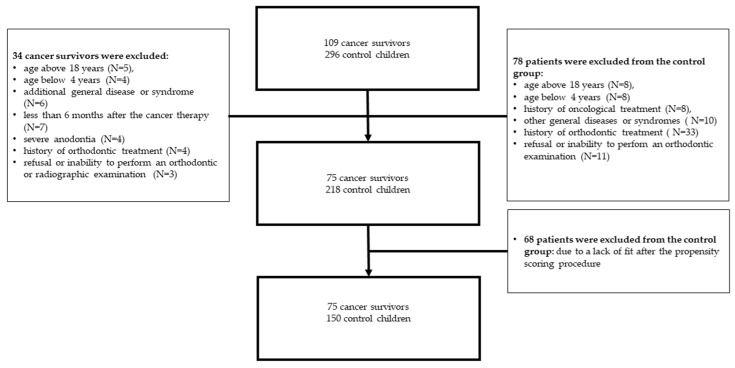
Diagram of inclusion and exclusion criteria for both groups.

**Figure 2 ijerph-19-04045-f002:**
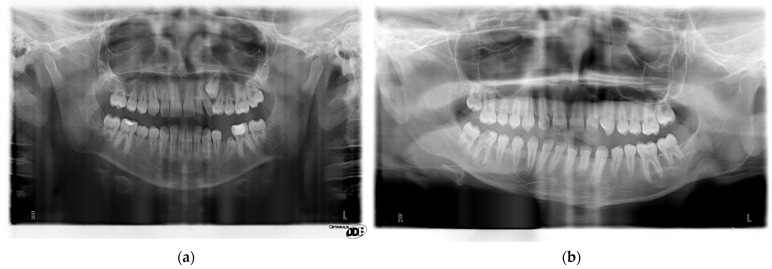
(**a**) Girl (cancer patient 1), aged 114 months, diagnosed with rhabdomyosarcoma (RMS) at the age of 57 months, treated for 36 months with CWS06-CEVAIE and TC protocols, with disease relapse, with posterior crossbite and persistent tooth 62, malformed roots and disturbed eruption of tooth 45; (**b**) Boy (cancer patient 2), aged 119 months, diagnosed with brain tumor at the age of 80 months, treated for 19 months with CZD II protocol and RTX of head, with anterior crossbite, spacing, and malformed roots.

**Table 1 ijerph-19-04045-t001:** Distribution of cancer diseases among patients.

Cancer Type	Girls	Boys	Total N (%)
Acute leukemia	9	25	34 (45.33%)
Wilms tumor	6	5	11 (14.66%)
Neuroblastoma	3	6	9 (12%)
Other soft tissue tumors	2	7	9 (12%)
Brain tumor	3	2	5 (6.66%)
Lymphoma	0	4	4 (5.33%)
Hepatoblastoma	0	3	3 (4%)
Total	23	52	75 (100%)

**Table 2 ijerph-19-04045-t002:** Distribution of dental occlusion types in cancer and control groups.

Occlusion	Cancer N (%)	Control N (%)	*p*
Malocclusion	49 (65.33%)	99 (65.56%)	0.9727
Class I	51 (68.00%)	94 (62.25%)	0.9361
Class II (distocclusion)	21 (28.00%)	52 (34.44%)	0.3670
Class III (reversed overjet)	6 (8.00%)	3 (1.99%)	0.0628
Abnormal overjet (incisor protrusion)	3 (4.00%)	5 (3.31%)	1.0000
Crossbite anterior	10 (13.33%)	6 (3.97%)	0.0136
Crossbite posterior	14 (18.67%)	7 (4.64%)	0.0012
Open bite anterior	1 (1.33%)	13 (8.61%)	0.0387
Open bite posterior	2 (2.67%)	0	0.1091
Deep bite	1 (1.33%)	12 (7.95%)	0.0651
Scissors bite	0	2 (1.32%)	1.0000
Crowding/rotations	6 (8.00%)	15 (9.93%)	0.8195
Malalignment	14 (18.67%)	12 (7.95%)	0.0310
Spacing	2 (2.67%)	0	0.1091
Midline shift/Mandibular displacement	3 (4.00%)	6 (3.97%)	1.0000
Maxillary narrowing	1 (1.33%)	11 (7.28)	0.0662

**Table 3 ijerph-19-04045-t003:** Distribution of tooth anomalies in cancer patients with different types of dental occlusion.

Occlusion		Agenesis of Teeth	*p*	Microdontic Teeth	*p*	Short Roots	*p*
Malocclusion	no	0 (0–2)	0.2160	0 (0–2)	0.7375	0 (0–0)	0.6838
	yes	0 (0–1)		0 (0–2)		0 (0–0)	
Class I	no	0 (0–1.5)	0.9061	0 (0–1.5)	0.8482	0 (0–0)	0.8479
	yes	0 (0–1)		0 (0–3)		0 (0–0)	
Class II (distocclusion)	no	0 (0–1)	0.9922	0 (0–3)	0.3874	0 (0–0)	0.8606
	yes	0 (0–1.5)		0 (0–1)		0 (0–0)	
Class III (reversed overjet)	no	0 (0–2)	0.2434	0 (0–2)	1.0000	0 (0–0)	0.5806
	yes	0 (0–0)		0 (0–4)		0 (0–0)	
Abnormal overjet (incisors protrusion)	no	0 (0-1)	0.9527	0 (0–2)	0.3433	0 (0–0)	0.2241
	yes	0 (0–3)		2 (0–4)		0 (0–2)	
Crossbite anterior	no	0 (0–2)	0.9093	0 (0–2)	0.7432	0 (0–0)	0.2143
	yes	0 (0–1)		0 (0–3.5)		0 (0–4)	
Crossbite posterior	no	0 (0–2)	0.6998	0 (0–2)	1.0000	0 (0–0)	0.0161
	yes	0 (0–1)		0 (0–1)		0 (0–2)	
Open bite posterior	no	0 (0–1.5)	0.3490	0 (0–2)	0.6288	0 (0–0)	0.6374
	yes	0 (0–0)		2 (0–4)		0 (0–0)	
Crowding/rotations	no	0 (0–2)	0.1725	0 (0–2)	0.7657	0 (0–0)	0.5152
	yes	0 (0–0)		0.5 (0–2.5)		0 (0–0)	
Malalignment	no	0 (0–1.5)	0.5333	0 (0–2)	0.3360	0 (0–0)	0.9388
	yes	0 (0–0)		1 (0–4)		0 (0–0)	
Midline shift/Mandibular displacement	no	0 (0–1)	0.0127	0 (0–2)	0.7617	0 (0–0)	0.5505
	yes	3 (1–7)		1 (0–1)		0 (0–0)	

Data with no evidence were excluded from the statistical analysis.

**Table 4 ijerph-19-04045-t004:** Comparison of dental age (DA-CA) among cancer patients with different types of dental occlusion.

Crossbite		Delta DA-CA	*p*
Class I	Yes	0.53 (0.13–1.27)	0.2611
	No	1.02 (0.17–2.13)	
Class II (distocclusion)	Yes	0.92 (0.13–1.65)	0.9936
	No	0.53 (0.33–1.75)	
Class III (reversed overjet)	Yes	0.92 (0.17–1.75)	0.4059
	No	0.28 (−1.27–1.83)	
Abnormal overjet (incisor protrusion)	Yes	0.92 (0.13–1.75)	0.9877
	No	0.53 (0.33–2.12)	
Crossbite anterior	Yes	1.60 (0.53–2.67)	0.2599
	No	0.84 (0.15–1.46)	
Crossbite posterior	Yes	1.75 (1.12–3.17)	0.0235
	No	0.80 (0.13–1.27)	
Crowding/rotations	Yes	0.93 (0.22–1.83)	0.4124
	No	0.53 (0.08–1.02)	
Malalignment	Yes	1.07 (0.31–1.98)	0.0226
	No	0.12 (−0.23–0.73)	
Spacing	Yes	0.91 (0.15–1.7)	0.6671
	No	1.7 (0.22–3.18)	

DA—dental age, CA—chronological age.

**Table 5 ijerph-19-04045-t005:** Cancer-related variables in patients with different types of malocclusion.

	Mesio-Distal Changes			Transversal Changes			Vertical Changes			Tooth Disturbances		
	Yes	No	*p*	Yes	No	*p*	Yes	No	*p*	Yes	No	*p*
	Median (25–75) percentile values)	Median (25–75) percentile values)		Median (25–75) percentile values)	Median (25–75) percentile values)		Median (25–75) percentile values)	Median (25–75) percentile values)		Median (25–75) percentile values)	Median (25–75) percentile values)	
Age at the beginning (mo)	33 (22–62)	33 (18.5–48.5)	0.4655	29.5 (17–52)	33 (22–54)	0.5450	31 (1–79)	33 (20–53)	0.7868	57 (28–80)	30.5 (17–39)	0.0198
Length of disease (mo)	13 (5–25)	21 (10–26)	0.2866	21.5 (6–26)	20 (6–25)	0.4839	24 (5–28)	20 (6–25)	0.7043	20 (13–26)	20.5 (6–25)	0.8342
RTX of head (yes)	3/23 *	8/52	1.00	4/22	7/53	0.7213	0/3	11/72	1.00	2/17	9/58	1.00
Relapse (yes)	2/23	6/52	1.00	4/22	4/52	0.2271	0/3	8/72	1.00	2/16	6/58	1.00
TBI (yes)	0/23	1/52	1.00	0/22	1/53	1.00	0/3	1/72	1.00	0/17	1/58	1.00

* Patients with RTX and malocclusion vs. patients with RTX only, RTX—radiotherapy of head or neck; TBI—total body irradiation.

## Data Availability

The data is available from the contact author on a reasonable inquiry.

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
