# Peer review of "Comparative Study of Malocclusions between Cancer Patients and Healthy Peers"

_ijerph, 2022, doi:10.3390/ijerph19074045_

Round 1

Reviewer 1 Report

The study investigates the prevalence of malocclusion among children survivors of a cancer diagnosis. The approach is very interesting, and although the children recruited in the study group are in a limited number as well as a low group homogeneity, the information provided might be the base for further research.

There are, however, some aspects to be modified:

  1. The abstract should be structured into a Background. Methods. Results and Conclusion.
  2. In the Introduction section, there are cited very old references, and the definitions/theories provided are not conclusive for this research in particular: the definition of normal occlusion and malocclusion should be rephrased also by citing more recent papers.
  3. It would also be interesting to providing more information about what normal occlusal relationships means in the case of children, in different age groups.
  4. Another issue regarding normal occlusal relationships would be to rethink and rephrase the statement ". More recently, in
    1981, Roth proposed that the definition should be based on the centric relation [3]." (doi: 10.1080/08869634.2021.1980685 )
  5. While centric relation is a great reference point in the case of extensive restorations in prosthodontics, or when temporomandibular disorders are diagnosed, maximum intercuspation cannot be neglected in the case of dentate patients.
  6. I believe that the Introduction section should be modified in this context, with much more recent citations.

Author Response

Open Review2

Dear Sir or Madam,

Thank you for your review. We are very thankful for all your comments that enhanced the quality of manuscript. All your suggestions were addressed accordingly and marked in the text. We hope you will find it satisfactory.

Your sincerely,

Patrycja Proc

Comments and Suggestions for Authors

The study investigates the prevalence of malocclusion among children survivors of a cancer diagnosis. The approach is very interesting, and although the children recruited in the study group are in a limited number as well as a low group homogeneity, the information provided might be the base for further research.

There are, however, some aspects to be modified:

The abstract should be structured into a Background. Methods. Results and Conclusion.

R. The abstract was corrected.

In the Introduction section, there are cited very old references, and the definitions/theories provided are not conclusive for this research in particular: the definition of normal occlusion and malocclusion should be rephrased also by citing more recent papers.

R. The changes have been introduced.

It would also be interesting to providing more information about what normal occlusal relationships means in the case of children, in different age groups.

R. Some more information about the norms of occlusion were given briefly. The subject of development of the dentition is very vast and the authors tried to introduce the general knowledge about it.

Another issue regarding normal occlusal relationships would be to rethink and rephrase the statement ". More recently, in 1981, Roth proposed that the definition should be based on the centric relation [3]." (doi: 10.1080/08869634.2021.1980685 )

While centric relation is a great reference point in the case of extensive restorations in prosthodontics, or when temporomandibular disorders are diagnosed, maximum intercuspation cannot be neglected in the case of dentate patients.

R: Thank you for the remark, the statement was removed. We believe that our study was carried out in the central occlusion, i.e. in the maximum habitual occlusion of the teeth, and not in the central relation, i.e. the ligamentous-muscular relationship, where the upper-anterior location of the condyles is taken into account. A relevant statement on this is contained in material and methods section.

I believe that the Introduction section should be modified in this context, with much more recent citations.

R. Thank you for the remark, the introduction was modified according to your recommendations.

Reviewer 2 Report

Dear authors,
First of all I would like to congratulate the authors for the present work and effort made. However, I would like to send you some considerations to take into account.
- The title is somewhat confusing. The prevalence of malocclusion in cancer patients should be done with larger sample sizes.In my opinion the title should be more focused on a "comparative study of malocclusions between cancer patients and healthy patients".

In the classification of malocclusion, class I, II and III dentition must be specified. Why teleradiography was not performed in patients for skeletal malocclusion.

Patients with developmental diseases usually have hypodontia and agenesis, especially rare diseases, and this has been scientifically proven.
However, in order to conclude that the patient has a greater tendency to vertical malocclusion, a classification by groups should be made comparing dolichofacials and brachyfacials, otherwise the bias in the results obtained is evident. It should therefore modify its results and limitations of the study.

Best Regards

Author Response

Open Review3

Dear Sir or Madam,

Thank you for your review. We are very thankful for all your comments that enhanced the quality of manuscript. All your suggestions were addressed accordingly and marked in the text. We hope you will find it satisfactory.

Your sincerely,

Patrycja Proc

Comments and Suggestions for Authors

Dear authors,

First of all I would like to congratulate the authors for the present work and effort made. However, I would like to send you some considerations to take into account.

- The title is somewhat confusing. The prevalence of malocclusion in cancer patients should be done with larger sample sizes. In my opinion the title should be more focused on a "comparative study of malocclusions between cancer patients and healthy patients".

R: Thank you for your comment. The title has been changed to : "Comparative study of malocclusions between cancer patients and healthy peers".

In the classification of malocclusion, class I, II and III dentition must be specified.

R: These were added in the introduction section.

Why teleradiography was not performed in patients for skeletal malocclusion.-

R: Radiographs were not performed on school children selected for comparison. This group was selected on a blind basis to avoid bias.

Patients with developmental diseases usually have hypodontia and agenesis, especially rare diseases, and this has been scientifically proven.

R: Thank you for the comment. We have evaluated this relationship and this finding was an issue of our previous work: “Dental Anomalies as Late Adverse Effect among Young Children Treated for Cancer” by Patrycja Proc, Joanna Szczepańska, Adam Skiba, Małgorzata Zubowska, Wojciech Fendler, Wojciech Młynarski, Cancer Res Treat. 2016;48(2):658-667.   DOI: https://doi.org/10.4143/crt.2015.193

However, in order to conclude that the patient has a greater tendency to vertical malocclusion, a classification by groups should be made comparing dolichofacials and brachyfacials, otherwise the bias in the results obtained is evident. It should therefore modify its results and limitations of the study.

R: The limitations and conclusions have been modified.

Round 2

Reviewer 1 Report

The modifications have been performed. Congratulation on the research!